# Assigning the Absolute Configuration of Inositol Poly- and Pyrophosphates by NMR Using a Single Chiral Solvating Agent

**DOI:** 10.3390/biom13071150

**Published:** 2023-07-19

**Authors:** Kevin Ritter, Nikolaus Jork, Anne-Sophie Unmüßig, Maja Köhn, Henning J. Jessen

**Affiliations:** 1Institute of Organic Chemistry, University of Freiburg, 79104 Freiburg, Germany; kevin.ritter@oc.uni-freiburg.de (K.R.); nikolaus.jork@ocbc.uni-freiburg.de (N.J.); a.unmuessig@googlemail.com (A.-S.U.); 2CIBSS—Centre for Integrative Biological Signalling Studies, University of Freiburg, 79104 Freiburg, Germany; maja.koehn@bioss.uni-freiburg.de; 3Institute of Biology 3, Faculty of Biology, University of Freiburg, 79104 Freiburg, Germany; 4BIOSS—Centre for Biological Signalling Studies, University of Freiburg, 79104 Freiburg, Germany

**Keywords:** InsPs, inositol polyphosphates, inositol pyrophosphates, chiral solvating agent, NMR, enantiomer assignment

## Abstract

Inositol phosphates constitute a family of highly charged messenger molecules that play diverse roles in cellular processes. The various phosphorylation patterns they exhibit give rise to a vast array of different compounds. To fully comprehend the biological interconnections, the precise molecular identification of each compound is crucial. Since the *myo*-inositol scaffold possesses an internal mirror plane, enantiomeric pairs can be formed. Most commonly employed methods for analyzing InsPs have been geared towards resolving regioisomers, but they have not been capable of resolving enantiomers. In this study, we present a general approach for enantiomer assignment using NMR measurements. To achieve this goal, we used ^31^P-NMR in the presence of L-arginine amide as a chiral solvating agent, which enables the differentiation of enantiomers. Using chemically synthesized standard compounds allows for an unambiguous assignment of the enantiomers. This method was applied to highly phosphorylated inositol pyrophosphates, as well as to lowly phosphorylated inositol phosphates and bisphosphonate analogs. Our method will facilitate the assignment of biologically relevant isomers when isolating naturally occurring compounds from biological specimens.

## 1. Introduction

Inositol phosphates are a family of cellular messenger molecules involved in numerous processes in eukaryotic cells [1,2,3,4,5,6,7]. The highly charged molecules constitute a sophisticated signaling network and are associated with important functions such as energy and phosphate homeostasis as well as exocytotic events. The molecular scaffold of these messengers is *myo*-inositol (**1**), a cyclohexane hexa-ol structure with a distinct 3D arrangement of the groups. A generally applied assignment of the positions at the ring originates from another subclass of inositol phosphates, the phosphatidyl inositol phosphates, where the glycerol backbone is attached to the 1 position of the inositol (numbering see Figure 1). The remaining positions are numbered counterclockwise. The mirror plane through the 2 and 5 positions of the molecule renders it a *meso*-compound, which is optically inactive and achiral. The introduction of groups outside of the mirror plane leads to a desymmetrization of the molecule and therefore to the formation of enantiomeric pairs. By combinatorial attachment of phosphate groups to the inositol ring, 63 different inositol phosphate derivatives are possible [8]. By introducing further phosphate groups to inositol hexakisphosphate (InsP_6_), forming phosphoanhydride bonds, inositol pyrophosphates (PP-InsPs) are formed. One additional phosphate group generates PP-InsP_5_ (also often abbreviated as InsP_7_). Six different PP-InsP_5_ isomers are possible, including two enantiomeric pairs (1/3-PP-InsP_5_ and 4/6-PP-InsP_5_), whereas the remaining two are achiral *meso*-compounds (Figure 1). By introducing another pyrophosphate group, 15 different (PP)_2_-InsP_4_ regioisomers (also often abbreviated as InsP_8_) are possible, including six pairs of enantiomers. Of note, both 1/3-PP-InsP_5_ and 4/6-PP-InsP_5_ have been identified in biological systems [9,10,11,12,13,14,15].

It becomes clear that there is a large number of different inositol phosphates depending on the number of phosphate groups and the positions where they are attached. To understand connections within the complex signaling network of inositol phosphates, accurate structural assignment is of great importance. In addition to the variety of different isomers, the absence of chromophores and low concentration in cells are major hurdles for accurate assignments. Different methods were established to analyze structures and concentrations of inositol phosphates even in a complex mixture or by previous enrichment. In the last few decades, the most commonly applied method relied on radiolabeling by using tritiated [^3^H]-inositol in combination with SAX-HPLC and subsequent scintillation counting. Gel electrophoresis (PAGE) is an alternative method that is inexpensive and readily available. Although it can separate higher InsPs, it has great difficulty distinguishing between PP-InsP regioisomers [16]. In 2020, we introduced the capillary electrophoresis electrospray ionization mass spectrometry (CE-ESI-MS) method combined with stable isotopic labeled standards, which enables accurate assignments without radiolabeling [17]. However, all of the presented methods have in common that they are only able to distinguish between regioisomers but not between enantiomers.

Another method for assigning biologically relevant isomers is based on NMR measurements. InsPs are isolated from enzymatic reactions or cell lysates, and in combination with chemically synthesized InsP standards, NMR spiking experiments are conducted [15,18,19,20]. But again, this method is not able to discriminate between enantiomers since the compounds are in an achiral environment and therefore the resonance signals of enantiomers are recorded at identical chemical shifts. A comparison of optical rotation values is not possible as those values are usually very small, are strongly pH-value-dependent, and require very high purity.

Yet, there are also methods that are capable of discriminating not only between regioisomers, but also between enantiomers. One method involves crystallization experiments. For instance, the naturally occurring (PP)_2_-InsP_4_ enantiomer was identified in crystal complexes of the kinase domain of human diphosphoinositol pentakisphosphate kinase 2 (PPIP5K2) [14,21]. Using a combination of enantiopure synthetic standards and enzyme assays, it is also possible to reveal the identity of biologically relevant isomers [22]. Recently the *Fiedler* group demonstrated the viability of enantiomer assignment by applying enantiopure [^13^C_1_]- and [^13^C_2_]-labeled *myo*-inositols in combination with ^13^C-, ^31^P-, and two-dimensional-NMR measurements [23].

Blüher et al. introduced an easy way to discriminate between InsP_5_ enantiomers using NMR spectroscopy by adding an enantiopure chiral solvation agent [24]. A chiral environment is generated, and the formation of diastereomeric ion pairs consisting of the respective InsPs and the shift reagent results in slightly different shifts for each enantiomer in ^31^P-NMR. By spiking with enantiopure standard compounds, an assignment is possible, yet it requires significant amounts of analyte on the order of 10–100 micrograms. Since it is known that phosphates interact strongly with guanidium groups, several compounds containing such groups were screened. It was found that commercially available L-arginine amide hydrochloride (**8**, L-ArgNH_2_ * HCl) is a suitable chiral solvating agent for 1/3-OH InsP_5_ (Figure 2) [24].

In this work, we extend this approach to PP-InsPs and to nonhydrolyzable PCP-InsP analogs that carry a bisphosphonate moiety (PCP) [25,26]. We also synthesized several model compounds to show the scope and limitations of the method concerning the phosphorylation density of the InsPs.

## 2. Materials and Methods

### 2.1. Materials

Enantiopure L-arginine amide hydrochloride (**8**) was purchased from Sigma Aldrich. Enantiopure PP-InsP_4_ and PP_2_-InsP_4_ were synthesized chemically using previously published procedures, with slight adjustments, e.g., pH values [21,22,27]. Enantiopure InsP_3_ and InsP_4_ standard compounds are commercially available and were obtained from SiChem as sodium salts. Description and synthesis of the compounds S1–S6 can be found within the Appendix A.

### 2.2. NMR Measurements

NMR measurements were conducted on a Bruker Ascend 400 spectrometer with an Avance Neo console equipped with a Prodigy Cryoprobe. ^31^P-NMR (162 MHz) spectra were acquired at 300 K and recorded with ^1^H-decoupling (pulse program: zgpg30; size of FID: 64k; NS: 320; acquisition time: 0.5 s; spectral width: 405 ppm).

All NMR spectra were acquired in D_2_O with a sample concentration ranging from 2 to 4 mm (approx. 1 mg/mL). To enhance the resolution of the resonance signals, all samples were supplemented with ~50 mm Na_2_EDTA. The amount of L-Arg-NH_2_ hydrochloride (**8**) added during the experiments was in the range of 100- to 150-fold excess and was adjusted accordingly. The pH value was adjusted by adding TFA and aq. NaOH solution.

## 3. Results

### 3.1. Chemical Synthesis of InsP Model Compounds

In order to test the method, different model compounds were synthesized. Racemic 1,4,5-InsP_3_ (**14**) was synthesized over eight steps (Figure 1). Starting from *myo*-inositol (**1**) different protecting groups were introduced to obtain the protected inositol **9** in accordance with the literature [28,29,30,31]. Subsequent protection of the 4-/6-position with an MEM group followed by selective cleavage of the silyl protecting groups gave triol **11**. Threefold phosphorylation, followed by sequential cleavage of all protecting groups, furnished *rac*-1,4,5-InsP_3_ (**14**) in an overall yield of 6%.

As an InsP_4_ model compound, racemic 1,4,5,6-InsP_4_ (**18**) was synthesized over four steps (Figure 2). In brief, ketalization of *myo*-inositol (**1**) furnished tetra-ol **15** in accordance with the literature [28]. Phosphorylation of tetra-ol **15** with AB-P-amidite **S1** (see Appendix A) afforded the AB-protected-InsP_4_ derivative **16.** Basic cleavage of the AB-protecting groups, followed by an acidic cleavage of the cyclohexylidene ketal, gave *rac*-1,4,5,6-InsP_4_ (**18**) in an overall yield of 18%.

To explore if the chiral solvating agent could also be used to distinguish bisphosphonates, a nonhydrolyzable 4-/6-PCP-InsP_5_ derivative (**23**) was synthesized (Figure 3) [25,26,32]. The 4- and 6-position of *myo*-inositol (**1**) is accessible by ortho-ester formation, followed by the selective introduction of a silyl protecting group to the 2-position. Monophosphorylation by using the PCP-phosphoramidite **S2** (see Appendix A) furnished the PCP derivative **20** as a mixture of diastereomers. Acidic deprotection and subsequent phosphorylation with an *o*-xylene-derived XE-phosphoramidite **S3** (see Appendix A) afforded the protected PCP-InsP_5_ derivative **21** [33]. Reaction control during the acidic deprotection was essential as the protected PCP group starts to migrate along the ring when the reaction time is too long. Global deprotection under hydrogenative conditions in the presence of a palladium catalyst furnished racemic 4-/6-PCP-InsP_5_ (**23**) after six steps in a yield of 5%.

Another novel compound that was synthesized in the context of this study is 5-PP-1/3-InsP_1_ (**28**). The molecule was made starting from *myo*-inositol (**1**) in nine steps (Figure 4). It may serve to assign potential inositol pyrophosphates originating from early intermediates in inositol phosphate metabolism [34]. The PMB-protected InsP_1_ derivative **24** was synthesized over five steps according to the literature [21]. In the next step, one of the remaining two OH-groups was phosphorylated with XE-phosphoramidite **S3** (see Appendix A), resulting in InsP_2_-derivative **25** as a racemic mixture [33]. Selective deprotection of the 5-phosphate was achieved by removing the Fm protecting groups under basic conditions. The pyrophosphate was generated by phosphorylation with Bn-phosphoramidite **S4** (see Appendix A) [35]. 5-PP-InsP_1_ (**28**) was obtained by cleaving the remaining protecting groups under hydrogenative conditions.

### 3.2. Enantiomer Assignment by ^31^P-NMR Using a Chiral Solvating Agent

As discussed above, *myo*-inositol (**1**) harbors a mirror plane through the 2/5 positions of the molecule. Depending on the substitution pattern, different sets of signals were obtained for the InsPs and PP-InsPs in the ^31^P{^1^H}-NMR spectrum. The monophosphate signals appear in the range from 2 to −2 ppm. Signals of the pyrophosphates are visible in the range of −8 to −12 ppm. The shifts are strongly dependent on the pH value and the counterions [36].

In order to extend our enantiomer discrimination approach for InsP_5_ enantiomers, we measured various InsP and PP-InsP compounds [24]. Starting with 1,3- and 3,5-PP_2_-InsP_4_ (compounds **S5** and **S6**, see Appendix A), the enantiopure compounds were mixed to obtain racemic or chiral non-racemic mixtures. Without the presence of L-ArgNH_2_*HCl (**8**), we observed the typical signal set in the ^31^P-{^1^H}-NMR spectrum (Figure 3a). As expected, four singlet peaks in the monophosphate region are visible, each of them with an integral of 1. The diphosphate peaks are not clearly resolved, but because of the coupling between the phosphorus atoms, each of them should occur as a doublet and have an integral of 1. In the absence of a chiral solvating agent, the enantiomers cannot be distinguished by ^31^P-{^1^H}-NMR due to enantiotopicity. The addition of L-ArgNH_2_*HCl (**8**) creates a chiral environment that leads to diastereotopicity and a slight observable shift (Δppm up to 0.05 ppm) in some of the signals of each analyte (Figure 3a). By preparing samples with different compositions, it is possible to assign the signals to each individual enantiomer. For example, in a 2:1 mixture, only one set of the peaks is increasing in relative intensity. Raising the concentration of the opposite enantiomer leads to a corresponding increase in the other set of peaks (Figure 3b). As a control experiment, we also measured the enantiopure compounds in the presence of L-ArgNH_2_*HCl (**8**), and no splitting of the signals was observed (Figure 3c).

As the method was successful in analyzing PP_2_-InsP_4_ enantiomers, it was further applied to the unsymmetrical PP-InsP_5_ compounds. Without chiral additive, 4- and 6-PP-InsP_5_ (**6** and **4**) showed identical resonances in the ^31^P-NMR spectrum when mixed together, and only one set of signals was observed (Figure 4a). However, upon addition of L-ArgNH_2_*HCl (**8**), we observed the splitting of resonance signals to distinct subsets, both in the monophosphate region and for the signals of the pyrophosphate group (Δppm up to 0.1 ppm). By spiking with enantiopure compounds, we are able to make an absolute assignment (Figure 4c). In a control experiment, the enantiopure compounds were also measured in the presence of L-ArgNH_2_*HCl (**8**), and—as expected—only one set of signals was observed. The experiment was also conducted with enantiopure 1- and 3-PP-InsP_5_ (**3** and **7**) and again could be applied successfully for an absolute assignment of enantiomer identity (Figure 5). Although not all signals were equally well resolved, an unambiguous assignment of the absolute configuration requires only one out of the many signals available. Although minor impurities are visible in the spectra (e.g., InsP_6_), they do not impede the assignment of the absolute configuration.

After demonstrating the method’s general applicability to inositols containing seven or eight phosphates, we aimed to evaluate it for compounds with lower degrees of phosphorylation compared to InsP_5_. Therefore, a racemic mixture of 1,4,5,6-InsP_4_ (**18**) was synthesized (see Figure 2) and ^31^P{^1^H}-NMR measurements were conducted. While the effectiveness of L-ArgNH_2_*HCl (**8**) was found to be largely independent of the pH value for InsP_5_, InsP_7_, and InsP_8_, a signal splitting into distinct subsets for InsP_4_ was only obtained within a narrow pH range. At a pH value between 7 and 8, we observed the expected splitting of the resonance signals (Δppm up to 0.06 ppm) in the presence of L-ArgNH_2_*HCl (**8**), whereas no splitting was observed in the absence of the chiral solvating agent (Figure 6a,b). By spiking with commercial enantiopure 1,4,5,6-InsP_4_ (**18a**), we were able to assign the signals to each respective enantiomer (Figure 6c). Outside of the pH range, the addition of the chiral solvating agent had no effect.

To further test the possibilities of the method regarding the number of phosphate groups, we conducted measurements with *rac*-1,4,5-InsP_3_ (**14**) under the same conditions mentioned above. The method was successful in resolving the enantiomer signals, but the pH adjustment was even more critical (Figure 7a). The desired splitting was only obtained at an exact pH of 6.4. The addition of the commercially available enantiomer allowed for the precise assignment of the respective signals in this case as well (Figure 7b).

After determining the limitations of the method with respect to the phosphorylation degree of InsPs, we also tested some non-natural InsP derivatives. One of them was 4-/6-PCP-InsP_5_ (**23**), which contains a non-hydrolyzable bisphosphonate (PCP) group substituting the diphosphate group. While these molecules do not occur in biology, we wanted to understand if the chiral solvating agent approach would also enable the determination of enantiomeric purity in such samples. Again, the addition of L-ArgNH_2_*HCl (**8**) resulted in the splitting of individual signals into subsets, independent of the pH value, allowing for the assignment of the respective enantiomers in principle (Figure 8a). However, we do not yet have an enantiomerically pure compound available to provide such an assignment. The other more unconventional InsP compound that we tested was the so far not naturally detected inositol pyrophosphate 5-PP-1/3-InsP_1_ (**28**) with only three phosphate groups (Figure 4). Similarly to the previously observed trend with lowly phosphorylated InsPs, the method was highly pH-dependent, so the presence of a pyrophosphate group alone is not sufficient to render the method pH-independent. A signal splitting was only obtained within a narrow pH range between 7 and 8, and the peak-to-peak resolution was small, only observable for the phosphate ester peak at 3.4 ppm (Figure 8b). This will render the assignment of such putative isomers from biological samples difficult. Due to the unavailability of enantiomerically pure compounds, we are unable to provide a definitive assignment for **28**.

## 4. Discussion

The accurate analysis and identification of regioisomers and enantiomers is a significant challenge but crucial for understanding the complex network of InsPs and PP-InsPs in biological systems [9,23,37,38]. Enantiomer assignment has traditionally been accomplished using crystal structures, enzymatic assays in combination with chemically synthesized standard compounds, and isotopically labeled *myo*-inositol (**1**) in combination with ^13^C-, ^31^P-, and two-dimensional-NMR measurements [14,15,18,19,20,21,22,23]. Here, we have demonstrated that assigning enantiomers can be achieved by using L-ArgNH_2_*HCl (**8**) as a chiral solvating agent in combination with ^31^P-NMR measurements. The method previously published for analyzing InsP_5_ enantiomers [24] has been extended to include highly phosphorylated PP-InsPs, lowly phosphorylated InsPs, and uncommon PCP-InsPs. The method showed excellent performance for the highly phosphorylated PP-InsPs without requiring any pH adjustment. However, for the lowly phosphorylated InsPs, pH adjustment during the measurement was necessary to improve resolution.

Our NMR-based method for enantiomer assignment can be used to assign the biologically relevant enantiomers of InsPs and PP-InsPs isolated from cells using, e.g., the TiO_2_ method and purified by SAX column chromatography or PAGE [39]. However, a significant amount of material is required due to the inherently low sensitivity of NMR spectroscopy, which can be a challenge in practice. In our earlier example of 1/3-OH InsP_5_, ca. 10 micrograms of material purified by PAGE was required for unambiguous enantiomer assignment. While this is a significant amount to isolate from biological samples, we and others have demonstrated that it is possible. Furthermore, the enantiomerically pure standard compounds need to be synthetically accessible, underscoring the importance of chemical synthesis in this interdisciplinary research area. An obvious advantage of the method described herein is that it relies on commercially available and cheap L-ArgNH_2_*HCl (**8**) that enables the resolution of enantiomeric pairs across a wide range of different inositol phosphorylation, pyrophosphorylation, and diphosphonate patterns. It is, therefore, a valuable addition to the current repertoire available to resolve the enantiomeric identity of InsPs and PP-InsPs.

## Data Availability

Additional data are available in the supplementary information and upon request from the authors.

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
