# Peer review of "Assigning the Absolute Configuration of Inositol Poly- and Pyrophosphates by NMR Using a Single Chiral Solvating Agent"

_biomolecules, 2023, doi:10.3390/biom13071150_

Round 1

Reviewer 1 Report

The authors have developed a method based on 31P NMR spectroscopy to perform the chiral recognition of inositol P-derivatives. The method employs L-arginine amide as a chiral solvating agent. The NMR protocol is sensitive to pH and requires a great amount of CSA, that is justified by the complex structure of the samples. The work provides different compound models and a good quality of NMR analyses.

I only suggest adding some chemical shift differences values (∆ppm) in the main discussion for each inositol evaluated which is important for a comparative visualization of the results from distinct compound models.

Author Response

We thank the reviewer for these very constructive comments. 

we have added some chemical shift differences in the text, indicating maximal values that were achieved. 

Reviewer 2 Report

The present report describes a method based on 31P NMR to characterize and assign enantiomers of inositol phosphates and pyrophosphates. The former act as signaling biomolecules in various cellular signal transduction pathways, with key roles in different vital processes. More recently discovered, the latter are still investigated to better understand their roles. To do so, it is crucial to be able to determine the structure of these inositol (pyro)phosphate derivatives and to be able to recognize their relative and absolute stereochemistry. 

Within a biological study, these authors already developed an NMR method able to identify an inositol phosphate. The same procedure was applied here and extended to discriminate enantiomers of various inositol (pyro)phosphates, and for that, new derivatives were prepared and fully characterized. Nevertheless, it may not be applied to all kind of inositol phosphate derivatives due to structural and pH limitation.

Although clearly not novel, this work offers an interesting tool, complementary to others, for specialists working in the topic of inositol phosphate and their biological effects. For this reason, it is worth publishing it.

Before publication, some points need to be clarified:

- p.1, it is written that ’…the glycerol backbone is attached to the 1 position of the inositol ‘ with a link to Fig. 1. However, the corresponding structure is not represented in this Figure. Please add the structure of this glycerophosphate for a better understanding.

- p. 4,5, it is not clear why some phosphoramidite reagents have been coined S1-3 (but not the PCP-phosphoramidite). Why S ? Furthermore, Sch. 3 mentioned the S1 reagent in step c, while it should be the XE-derivative S3.

the term AB-P amidite need to be explicated in the text.

the terms PCP and XE need to be explicated in the text.

The compound named PCP-InsP5 exhibit the number 24 in the text, but 23 in Sch. 3.

Note that the usual abbreviation for the fluorenylmethyloxycarbonyl protecting group is Fmoc, not Fm (Sch. 4).

- p. 6-7, 1,3- and 3,5-PP2-InsP4 are now labeled S5 and S6, although they are not phosphoramidite reagents (cf above). This is quite confusing.

A few typos could also be found:

- p3, line 86, labeled instead of ‘labled’ 

- p. 6, other numbering problems appear in Sch. 4; compounds labeled 25 and 27 in the caption must be 24 and 26.

Author Response

We thank the reviewer for the very constructive comments. 

We have added numbers in figure 1 (and mention this in the text). We do not want to show the whole structure, because it is not part of the developed NMR method described herein. It is simply mentioned to introduce the numbering. 

We have added information that S numbering refers to structures shown in the SI. This was also requested by the other reviewer. 

AB, PCP and XE are explained in the figures. We have added explanations of the structures in the captions of the figures.  We have added text that explains the PCP structure (bisphosphonate moiety (PCP)). 

We have corrected the issues with the numbering. Thanks for catching this!

Fm is actually fluorenylmethyl. Fmoc is Fluorenylmethyloxycarbonyl, which we did not use here. 

We thank the reviewer for bringing up these issues. We hope that the corrected version will be acceptable. 

Reviewer 3 Report

The manuscript by Jessen and colleagues describes the structural resolution of the polyphosphate and pyrophosphate enantiomers of inositol by NMR in solution. In particular, the authors' work focuses on the use of an enantiopure chiral agent that acts as a ligand to form relatively stable complexes in solution that result in disatheromeric pairs with the racemic mixture of inositol derivatives.

In my opinion, I find the work very well done with enough experimental evidence to support the final conclusions. The agent chosen to act as ligand (L-arginine amide hydrochloride) proves to be adequate to resolve both diasteromers into two sets of signals and the spectra show that unambiguous assignment of each parent enantiomer is possible.

The work shold be published in the form in which it is written. I recommend the authors to correct some typographical errors that I point out below. Also, reading the manuscript, I had a hard time deducing the structures of compounds S1 - S6, until I realized that these compounds were described in the supplementary material. I suggest that the authors add for the compounds that appear throughout the manuscript as S1 - S6, the indication that their synthesis and description are found within the supplementary material, for example: S1 (see Suplementary Material).

Typographical error that shoud be corrected:

1) Page 5, line 153: Change "24" by "23"

2) Scheme 3: Change "PCP - 4/6-InsP7" by "PCP - 4/6-InsP5"

3) Scheme 4: Change "29" by "28"

4) Figure 4: Please revise the caption, a) and c) are missing and b) is described twice.

5) Page 12, line 331: Supplementary Materials: the content of this section shoud be described here.

Author Response

We thank the reviewer for the very constructive comments. 

We have added the requested changes. Thank you for picking up the typos!